# A Non-Specific Phytohormone Regulatory Network in *Saccharina japonica* Coordinates Growth and Environmental Adaptation

**DOI:** 10.3390/plants14121821

**Published:** 2025-06-13

**Authors:** Jiexin Cui, Jinli Zhu, Yinru Dai, Jincheng Yuan, Wen Lin, Tao Liu

**Affiliations:** 1State Key Laboratory of Marine Environmental Science, Xiamen University, Xiamen 361102, China; cuijiexin@stu.xmu.edu.cn (J.C.);; 2College of Ocean and Earth Sciences, Xiamen University, Xiamen 361102, China; 3Fisheries College, Jimei University, Xiamen 361021, China

**Keywords:** *Saccharina japonica*, phytohormones, metabolomics, transcriptomics, tissue-specific adaptation, abiotic stress

## Abstract

*Saccharina japonica* (*S. japonica*) is a large-scale intertidal aquatic plant that exhibits characteristics such as rhizoid, holdfast, and blade differentiation. It demonstrates remarkable environmental adaptability. However, compared with higher plants, details about its phytohormone content, distribution, synthesis, and accumulation remain poorly understood. In this study, the phytohormone contents distribution and expression patterns of synthetic genes in different parts of *S. japonica*, including the rhizoid, petiole, basis, middle, and tip, were analyzed in detail by combining targeted metabolomics and transcriptomics analyses. A total of 20 phytohormones were detected in *S. japonica*, including auxin, abscisic acid (ABA), cytokinin (CTK), ethylene (ETH), gibberellin (GA), jasmonate acid (JA), and salicylic acid (SA), with significant site-differentiated accumulation. ABA and JA were significantly enriched in the tips (28.01 ng·g^−1^ FW and 170.67 ng·g^−1^ FW, respectively), whereas SA accumulated specifically only in the rhizoid. We also identified 12 phytohormones, such as gibberellin A1, methyl jasmonate, and trans-zeatin for the first time in *S. japonica*. Transcriptomic profiling revealed the tissue-specific expression of phytohormone biosynthesis genes, such as *CYP735A* (CTK synthesis), in the rhizoids and *LOX/NCED* (JA/ABA synthesis) in the tips. Key pathways, such as carotenoid biosynthesis and cysteine methionine metabolism, were found to be differentially enriched across tissues, aligning with hormone accumulation patterns. Additionally, an enrichment analysis of differentially expressed genes between various parts indicated that different parts of *S. japonica* performed distinct functions even though it does not have organ differentiation. This study is the first to uncover the distribution characteristics of phytohormones and their synthetic differences in different parts of *S. japonica* and elucidates how *S. japonica* achieves functional specialization through non-specific phytohormone regulation despite lacking organ differentiation, which provides an important theoretical basis for research on the developmental biology of macroalgae and their mechanisms of response to adversity.

## 1. Introduction

*Saccharina japonica* (*S. japonica*), an intertidal cold-water brown algae endemic to the Pacific Northwest coast, is a subduction zone algae [1]. It is classified as Heterokontophyta, Phaeophyceae, Laminariales, and Laminariaceae [2]. *S. japonica* is one of the most important large-scale economic macroalgae in China, ranking first in the world in terms of its cultivated area and production, and is mainly cultivated in Liaoning, Shandong, Fujian, and other places [3,4]. As a foundation species, *S. japonica* forms extensive underwater forests that provide critical habitats for marine biodiversity while contributing substantially to carbon sequestration through its high primary productivity [5]. The commercial cultivation of *S. japonica* exceeds eight million tons annually (FAO, 2024), serving as a vital resource for alginate extraction, functional food production, and bioenergy feedstocks [6]. In addition, *S. japonica* has an alternating life history of heteromorphic generations consisting of sporophyte generations of multicellular chloroplasts and gametophyte generations of unicellular and uniseriate cellular filoplasts, which is of special research value in the study of the evolution of multicellular algae [7]. Despite *S. japonica*’s ecological dominance and industrial significance, the molecular mechanisms regulating its developmental differences and environmental adaptation remain poorly understood in comparison to those of terrestrial plants.

Phytohormones are a class of organic compounds produced by plants, which play important fundamental regulatory roles in growth and development [8]. Nine main groups of plant hormones have been reported to date, namely, abscisic acid (ABA), auxin (Aux/IAA), brassinosteroids (BR), cytokinin (CTK), ethylene (ETH), gibberellin (GA), jasmonate acid (JA), salicylic acid (SA), and strigolactone (SL) [9,10]. Phytohormones can induce positive growth effects on treated plants in very small concentrations [11]. They are involved in the regulation of plant rooting, germination, flowering, fruiting, dormancy, and other life processes and have complex physiological effects on the division, differentiation, and elongation of plant cells [12,13]. Algae share many similarities with higher plants regarding phytohormones [14,15]. For example, *Sargassum horneri* contains large amounts of phytohormones, including Aux/IAA, indolebutyric acid (IBA), ABA, and zeatin (ZT) [16] and *Cladophora glomerata* contains phytohormones, such as IAA, IBA, and 6-benzylaminopurine (6-BA) [17]. Researchers have also identified IAA, trans-zeatin-riboside (tZT), ABA, and SA in *Pachymenia carnosa*, *Chondrus ocellatus*, and *Ulva Lactuca* [18]. Phytohormones also play important roles in algal growth, development, and resistance to adversity stress. Researchers have found that auxin not only promotes algal cell growth and division but also regulates the accumulation of proteins, photosynthetic pigments, and lipid metabolites and affects the antioxidant capacity of algae [19]. CTK has been demonstrated to enhance the proliferation of cells, as well as the levels of chlorophyll, carotenoids, monosaccharides, and glycolic acid in *Chlorella vulgaris* [20]. ABA promotes algal growth and may help algae resist adversity stress [21]. The effects of phytohormones on algal growth, development, and their distribution within algae are relatively well understood. However, knowledge regarding the biosynthesis, the metabolic pathways of algal phytohormones, and the mechanisms underlying hormone interactions remains limited.

Macroalgae such as *S. japonica* do not present organ differentiation but possess complex structures, with only blades, petioles, and holdfasts present, lacking the organ differentiation seen in higher plants [22,23]. Despite the distinct morphological characteristics exhibited by these different tissue regions of *S. japonica*—for example, the holdfast being root-like (rhizoidal), the petiole being cylindrical, and the blade being strap-shaped—they all share the same fundamental tissue organization, consisting of the epidermis, cortex, and medulla. Studies have shown that the phytohormones detected in *S. japonica* are auxin, CTK, ABA, GA, SA, and JA [18,24], which change with the growth period [25]. However, while previous research has primarily focused on how environmental factors such as high temperatures affect phytohormonal levels in *S. japonica* [26], studies investigating the mechanism of the synthetic accumulation of these hormones and whether they are transported between various parts of *S. japonica* are still limited. For *S. japonica*, their photosynthetic pigments are ubiquitously distributed throughout all epidermal cells of the thallus [27]. This indicates an absence of the functional differentiation of organs such as plants, for example, with leaves serving as primary photosynthetic organs and rhizoids functioning as water/nutrient absorption organs in higher plants [28,29]. This condition reflects the relatively primitive tissue/organ differentiation in macroalgae such as *S. japonica*, which exhibit morphological distinctions without functional specialization. However, for these original-type “foliose thallus” macroalgae, a paucity of metabolic studies remains in addressing potential functional differentiation across distinct tissue regions.

Understanding how phytohormones are differentially synthesized in the rhizoid (R), petiole (P), basis (B), middle (M), and tip (T) could provide deeper insights into the adaptive mechanisms of *S. japonica*. By combining targeted metabolomic and transcriptomic analyses, we can obtain a more comprehensive understanding of these processes. Targeted metabolomics will help elucidate the differential accumulation of phytohormones in different parts of *S. japonica*, while transcriptomics can offer valuable insights into the regulation of gene expression and the differences in synthesis. The combination of these two approaches will not only reveal the differences in phytohormone synthesis across the parts of *S. japonica* but also clarify the biological significance of these differences and their roles in the plant’s growth, adaptation, and environmental response.

## 2. Materials and Methods

### 2.1. Sample Collection

*Saccharina japonica* sporophytes (Sanhai, cultivated species, 150–200 cm) were collected from Xiapu, Fujian Province, China (119°92′ E, 26°56′ N), in late March 2024. Five parts of *S. japonica* were selected, the rhizoid (R), petiole (P), basis (B), middle (M), and tip (T) (Figure 1), with three biological replicates for each part sampled. After harvesting, the samples were washed and wiped clean, immediately frozen in liquid nitrogen, and placed in a −80 °C cryogenic refrigerator for storage.

### 2.2. Metabolite Sample Extraction

Methanol, acetonitrile, and acetic acid were purchased from ANPEL (Shanghai, China). All solvents were of LC–MS grade. And ultra-pure water in-house was prepared using a Milli-Q water purification system (Millipore, Bedford, MA, USA). Standards were acquired from Sigma-Aldrich (St. Louis, MO, USA). Standard stock solutions were produced at 10 mg/mL concentration in 50% MeOH. All stock solutions were kept at −20 °C. Before analysis, the stock solutions were diluted with 50% MeOH to generate working solutions.

Metabolite extraction was performed using a previously published [30,31] and modified method. Approximately 100 mg of frozen *S. japonica* material samples were extracted in 1 mL of ice-cold 50% aqueous acetonitrile (*v*/*v*). The sample was placed on ice, sonicated (30 Hz, 3 min) using a Portable Ultrasonic Homogenizer (Ningbo Scientz Biotechnology, Ningbo, China), and then extracted using a benchtop laboratory rotator at 15 rpm and 4 °C (performed in a 4 °C cold room) for 30 min. After centrifugation at 12,000 rpm for 10 min at 4 °C, the supernatant was transferred to clean plastic microtubes. All samples were purified using an ACQUITY UPLC HSS T3 (1.8 μm, 2.1 mm × 50 mm) (Waters, Milford, MA, USA), polymer-based, solid phase extraction (RP-SPE) cartridge, that had been washed with 1 mL of methanol and 1 mL of deionized water, then equilibrated with 50% aqueous acetonitrile (*v*/*v*). After loading a sample, the cartridge was then rinsed with 1 mL of 30% acetonitrile (*v*/*v*), and this fraction was collected. After this single-step SPE, the samples were evaporated to dryness under a gentle stream of nitrogen and stored at −20 °C until analysis. For the UHPLC-ESI-MS/MS analysis, the samples were dissolved in 200 μL of 30% acetonitrile (*v*/*v*) and transferred to insert-equipped vials.

### 2.3. UPLC Conditions and LC-MS/MS Analysis

The sample extracts were analyzed using an UPLC-Orbitrap-MS system (UPLC, Vanquish; MS, QE) (Thermo Fisher, Waltham, MA, USA). The analytical conditions were as follows: (1) column type, Waters ACQUITY UPLC HSS T3 (1.8 μm, 2.1 mm × 50 mm) (Waters, Milford, MA, USA); (2) column temperature, 40 °C; (3) flow rate, 0.3 mL/min; (4) injection volume, 2 μL; (5) mobile phase, phase A with ultrapure water (containing 0.1% acetic acid) and phase B with acetonitrile (containing 0.1% acetic acid); and (6) elution gradient, 0 min water/acetonitrile (85:15 *v*/*v*), 0.5 min water/acetonitrile (85:15 *v*/*v*), 1.5 min water/acetonitrile (10:90 *v*/*v*), 3 min water/acetonitrile (10:90 *v*/*v*), 3.1 min water/acetonitrile (85:15 *v*/*v*), and 5 min water/acetonitrile (85:15 *v*/*v*). To avoid the effects of fluctuations in the instrumental detection signal, the samples were analyzed continuously in a randomized order. Quality control (QC) samples were inserted into the sample queue to monitor and evaluate the stability of the system and the reliability of the experimental data.

HRMS data were recorded on a Q Exactive hybrid Q-Orbitrap mass spectrometer equipped with a heated ESI source (Thermo Fisher, Waltham, MA, USA) utilizing the SIM MS acquisition methods. The ESI source parameters were set as follows: spray voltage, 3.0 kV; sheath gas pressure, 40 arb; aux gas pressure, 10 arb; sweep gas pressure, 0 arb; capillary temperature, 320 °C; and aux gas heater temperature, 350 °C. Data were acquired on the Q-Exactive using Xcalibur 4.1 (Thermo Fisher, Waltham, MA, USA), and processed using TraceFinder™ 4.1 Clinical (Thermo Fisher, Waltham, MA, USA). Quantified data were output into excel format.

### 2.4. Total RNA Extraction

The SteadyPure RNA Extraction Kit (Accurate Biotechnology, Changsha, China) was used for the extraction and purification of total RNA from the plant tissue according to the manual instructions. Approximately 80 mg of tissue samples was ground to a powder within liquid nitrogen and then transferred to a 1.5 mL centrifuge tube with an AG RNAexPro Reagent. After incubation at room temperature for 5 min, the samples were centrifuged at 12,000× *g* for 5 min at 4 °C. Then, the supernatant was transferred to a new 1.5 mL centrifuge tube and subsequent purification was carried out according to the steps in the instructions. Subsequently, the total RNA was qualified and quantified using a Nano Drop and Agilent 2100 bioanalyzer (Thermo Fisher, Waltham, MA, USA).

### 2.5. RNA-Seq

After integrity and quality were verified, the RNA was subjected to cDNA library construction. RNA sequencing was executed using the NovaSeq 6000 platform (lllumina Inc., San Diego, CA, USA) by Beijing Annoroad Gene Technology Co., Ltd. (Beijing, China). The raw data were deposited in the Sequence Read Archive (SRA) database of NCBI with the BioProject (accession no. PRJNA1244720). Quality control checks were conducted using SOAPnuke v1.5.2. The clean reads were separated from the raw data by removing adaptor sequences, reads with more than 5% of unknown bases, and low-quality reads (the ratio of bases with a quality value less than 10 to total bases was more than 20%). The expression levels of each unigene were quantified with the RSEM software package v1.3.1 and presented as fragments per kilobase million (FPKM).

### 2.6. Identification of Differentially Expressed Genes (DEGs)

A differential expression analysis between two comparison combinations was performed using DESeq (1.20.0). Differentially DEGs were screened under the conditions of expression difference ploidy|log2(FoldChange)| > 1 and significance *p*-value < 0.05. An enrichment analysis was performed based on hypergeometric testing, a hypergeometric distribution test of KEGG was performed on pathway units, and genetic annotation was performed [32]. Using GO-Term in the GO database, a hypergeometric test of GO was performed to find GO-Terms, which was significantly enriched in DEGs compared with the entire genome background [33].

### 2.7. Real-Time Fluorescent Quantitative Analysis

Total RNA from *S. japonica* was extracted using the HP Total RNA Kit (Omega Bio-Tek, Norcross, GA, USA), and the first strand cDNA was synthesized using the TransScript^®^ All-in-One First-Strand cDNA Synthesis SuperMix for qPCR (One-Step gDNA Removal) (TransGen Biotech, Beijing, China). The cDNA was then diluted three times to serve as a template for real-time fluorescent quantitative PCR (qRT-PCR), with EF1α as the reference gene. The qRT-PCR reagents used were PerfectStart^®^ Green qPCR SuperMix (TransGen Biotech, Beijing, China). Based on the transcriptome data under *S. japonica* stress conditions, the sequences of the genes with the most pronounced up-regulation of expression in each gene were screened for transcriptional studies, and the nucleic acid sequences of the target molecules were obtained from the *S. japonica* genome data (unpublished). Primers required for the qRT-PCR experiment were designed using Primer Premier 5 software (Premier Biosoft, Palo Alto, CA, USA), and their sequences are shown in Table 1. The 2^−∆∆Ct^ method was used to calculate the relative transcription levels of the genes.

### 2.8. Data Analysis

Data collection and preliminary processing were conducted using Microsoft Excel 2021 software (Microsoft Corporation, Redmond, WA, USA). The statistical analysis of the experimental data was performed using IBM SPSS Statistics 27 software (IBM Corporation, Armonk, NY, USA). To assess the significant differences in photosynthetic parameters and physiological indicators among various treatment groups, an analysis of variance (ANOVA) was initially conducted, followed by Duncan’s multiple range test to elucidate the disparities between groups. In the presentation of results, distinct letters indicate significant differences between treatment groups at the same time point. A *p*-value of less than 0.05 was considered statistically significant. Graphs and charts were generated using GraphPad Prism 10 software (GraphPad Software, San Diego, CA, USA).

## 3. Results

### 3.1. Phytohormone Content Assays in Different Parts of S. japonica

A total of 24 phytohormones were detected from methanol extracts of different parts (R, P, B, M, and T) of *S. japonica* using HPLC-MS/MS (Table 2). A good coincidence was obviously observed in the chromatography peaks among three repeated samples in both modes (Appendix A). A Pearson correlation analysis further indicated good consistency in the repeated samples (85% coefficient > 0.83, Figure 2A). However, the coefficients between each pair of rhizoids, petioles, basis, middles, and tips were mostly less than 0.8. A principal components analysis (PCA) indicated good consistency within each group and significant differences between the groups (Figure 2B).

A total of 20 phytohormones spanning seven major classes were identified, including auxins (indole-3-acetic acid, 3-indolebutyric acid, 3-indolecarboxylic acid, 3-indolepropionic acid, 3-indoleacetonitrile, and 3-indoleacetamide), CTK (trans-zeatin, trans-zeatin-riboside, N6-(delta2-isopentenyl) adenine, N6-(delta2-isopentenyl) adenosine, and dihydrozeatin), ABA, GA (gibberellin A1 and gibberellin A4), SA, Eth (synthetic precursor, aminocyclopropane carboxylic acid), and JA (Figure 3, Appendix A). These phytohormones exhibited distinct tissue-specific distribution patterns in *S. japonica*, with the majority being ubiquitously detected across all five examined parts. Notably, specific compounds showed exclusive localization: 3-Indoleacetonitrile was uniquely identified in the tips, while SA demonstrated rhizoid-specific accumulation.

A detailed analysis was conducted on the phytohormones in *S. japonica* and the following findings were obtained (Figure 4): Auxin (indole-3-acetic acid) concentrations ranged from 0.641 to 2.585 ng·g^−1^ fresh weight (FW), with maximal accumulation in the tip followed by the middle. ABA content varied significantly among different parts, with a surge of as much as 28.013 ng·g^−1^ FW in the tip and only 0.476–1.595 ng·g^−1^ FW in other parts. CTK was low in *S. japonica*, but also varied between sites, with the tips of the blade having the highest content of 1.059 ng·g^−1^ FW. GA was also less abundant, with the highest content of 1.285 ng·g^−1^ FW in the tip and the lowest content of 0.193 ng·g^−1^ FW in the middle of the blade. The ETH precursor aminocyclopropane carboxylic acid accumulated predominantly in the rhizoid (4.215 ng·g^−1^ FW), showing nonsignificant differences among other regions. JA demonstrated a comparable profile to ABA. However, a highly significant difference was observed in the contents of the tip compared with that in the other parts. The contents were 29.62-, 91.50-, 12.51-, and 8.30-fold higher than those of the rhizoid, petiole, basis, and middle, respectively.

### 3.2. Quality Check of Transcriptome Sequencing Data

The total cDNA library prepared from different parts of *S. japonica* was sequenced using NovaSeq™ 6000 (lllumina Inc., San Diego, CA, USA), and the results are listed in Table 3. Quality information on the transcriptome sequencing of *S. japonica*. After cleaning and quality checking, 81,220,748 to 93,647,616 clean reads (1,309,838,908 in total), with Q30 base contents ranging from 93.94% to 94.72% and GC contents ranging from 54.78% to 56.01%, were generated from the cDNA libraries. The Pearson correlation analysis and PCA both indicated good consistency between repeated samples and significant differences among different parts of the transcriptomic data (Figure 5A,B).

### 3.3. Transcriptome Analysis of S. japonica Parts

The correlation heat map revealed a high correlation between different parts of *S. japonica*, indicating a positive correlation of gene expression among these parts (Figure 6A). A total of 14,938 genes were found to be co-expressed in all five parts, with 268, 185, 92, 106, and 69 genes uniquely expressed in the rhizoid, petiole, basis, middle, and tip of *S. japonica*, respectively (Figure 6B). These results show significant differences in gene expression between different parts of *S. japonica*.

Using the DESeq method, pairwise comparisons were conducted among five samples from *S. japonica*, namely the rhizoid, petiole, basis, middle, and tip. The criteria for screening differentially expressed genes (DEGs) were set at Qvalue < 0.05 or FDR ≤ 0.001 (Figure 6C). In the R vs. S comparison, a total of 3766 DEGs were detected, with 1888 up-regulated and 1878 down-regulated. In the R vs. B comparison, 3779 DEGs were identified, comprising 2078 up-regulated and 1701 down-regulated genes. The R vs. M comparison revealed 3836 DEGs, with 2248 up-regulated and 1588 down-regulated. The R vs. T comparison showed 6687 DEGs, including 3953 up-regulated and 2734 down-regulated genes. In the S vs. B comparison, 4182 DEGs were detected, with 2251 up-regulated and 1931 down-regulated. The S vs. M comparison identified 4872 DEGs, consisting of 2647 up-regulated and 2225 down-regulated genes. The S vs. T comparison revealed 6604 DEGs, with 3722 up-regulated and 2882 down-regulated. In the B vs. M comparison, 298 DEGs were detected, with 214 up-regulated and 84 down-regulated. The B vs. T comparison showed 6615 DEGs, including 3736 up-regulated and 2879 down-regulated genes. Finally, the M vs. T comparison revealed 5936 DEGs, with 3212 up-regulated and 2724 down-regulated. The number of DEGs in the R vs. T, S vs. T, and M vs. T comparisons were found to be similar and significantly higher than those in the other comparisons. These comparisons all focused on the tip (T) of *S. japonica*, indicating that the gene expression pattern in the tip is notably distinct from that in the other parts. The B vs. M comparison had the fewest DEGs, suggesting that the gene expression patterns in the basis and middle parts of *S. japonica* are relatively similar. Across all comparisons, the number of up-regulated genes was greater than the number of down-regulated genes.

The DEGs from the 10 comparison groups (R vs. P, R vs. B, R vs. M, R vs. T, P vs. B, P vs. M, P vs. T, B vs. M, B vs. T, and M vs. T) were analyzed for Gene Ontology (GO) functional annotation. The results revealed that the annotated DEGs across the 10 comparison groups could be classified into 3 major classes and 42 subclasses: biological processes (24 subclasses), molecular functions (13 subclasses), and cellular components (5 subclasses). Among these groups, the DEGs from the three comparison groups were relatively enriched in specific functional categories. In the biological process group, these included protein phosphorylation, signal transduction, proteolysis, and protein transport. In the molecular function group, the enriched categories were ATP binding, metal ion binding, and protein serine kinase activity. In the cellular component group, the DEGs were predominantly annotated to the cytoplasm and nucleus (Figure 7). The number of annotated DEGs were 1500 in the R vs. P comparison, 1309 in R vs. B, 1361 in R vs. M, and 2741 in R vs. T. This suggests that the metabolic changes in the rhizoid are relatively similar to those in the other parts (petiole, basis, and middle) but significantly different from those in the tip. In the comparison between P and B, 1735 DEGs were annotated, while 2051 were annotated in P vs. M and 2832 in P vs. T. This indicates that the metabolic changes in the petiole are similar to those in the basis but differ significantly from those in the middle and tip. In the comparison between B and M, 100 DEGs were annotated, while 2770 were annotated in B vs. T and 2483 in M vs. T. This suggests that the metabolic changes in the basis and middle are relatively similar but differ significantly from those in the tip. Additionally, substantial differences in metabolic changes compared with the tip were also evident in the middle part. These findings highlight distinct gene expression and metabolic patterns in the tip of *S. japonica* compared with in the other parts, while certain similarities exist among the rhizoid, petiole, basis, and middle parts.

The GO enrichment analysis revealed that the DEGs in each comparison group were significantly enriched in multiple GO terms. Specifically, the DEGs in the R vs. P, R vs. B, and B vs. M groups were primarily enriched in the cellular component (CC) “periplasmic space” (GO:0042597) and the molecular function (MF) “isomerase activity” (GO:0016853). In contrast, the R vs. M and P vs. M groups were enriched in “periplasmic space” (CC, GO:0042597) and “chlorophyll binding” (MF, GO:0016168). The R vs. T and B vs. T groups showed significant enrichment in the biological process (BP) “lipid metabolic process” (GO:0006629) and “chloroplast” (CC, GO:0009507), while the P vs. B and P vs. M groups were mainly enriched in “photosystem II” (CC, GO:0009523) and “structural constituent of ribosome” (MF, GO:0003735) or “chlorophyll binding” (MF, GO:0016168). The P vs. T group was enriched in “nucleolus” (CC, GO:0005730) and “preribosome, large subunit precursor” (CC, GO:0030687), whereas the M vs. T group was enriched in “proteasome complex” (CC, GO:0000502) and “alginic acid biosynthetic process” (BP, GO:0042121). Additionally, the B vs. T group was significantly enriched in “DNA-templated transcription termination” (BP, GO:0006353). The enrichment results for each group were visualized using bubble charts (Appendix A), highlighting the significant functional differences in gene expression among the different parts.

To further the functional characterization of the DEGs in each comparison group, we performed a pathway analysis based on the KEGG database. The KEGG pathway analysis of R vs. P (Figure 8A) revealed that the DEGs were mainly annotated into carbon fixation in photosynthetic organisms, carbon metabolism, and pentose phosphate pathway. R vs. B (Figure 8B) revealed that the DEGs were mainly annotated into carbon metabolism, peroxisome, and cysteine and methionine metabolism. R vs. M (Figure 8C) revealed that the DEGs were mainly annotated into carbon metabolism, biosynthesis of amino acids, and amino sugar and nucleotide sugar metabolism. R vs. T (Figure 8D) revealed that the DEGs were mainly annotated into proteasome, pentose and glucuronate interconversions, and carotenoid biosynthesis. P vs. B (Figure 8E) revealed that the DEGs were mainly annotated into ribosome, arginine and proline metabolism, and nitrogen metabolism. P vs. M (Figure 8F) revealed that the DEGs were mainly annotated into ribosome, nitrogen metabolism, and folate biosynthesis. P vs. T (Figure 8G) revealed that the DEGs were mainly annotated into ribosome biogenesis in eukaryotes, carbon metabolism, and cysteine and methionine metabolism. M vs. T (Figure 8H) revealed that the DEGs were mainly annotated into proteasome, pentose and glucuronate interconversions, and carbon metabolism. B vs. M (Figure 8I) revealed that the DEGs were mainly annotated into cysteine and methionine metabolism, MAPK signaling pathway–plant, and plant hormone signal transduction. B vs. T (Figure 8J) revealed that the DEGs were mainly annotated into carbon metabolism, pentose and glucuronate interconversions, and proteasome.

### 3.4. Expression Analysis of Phytohormone Synthesis-Related Genes in Different Parts of S. japonica

Phytohormone is one of the most important components involved in the growth and development of *S. japonica*, helping *S. japonica* to resist abiotic stress, and is mainly expressed in the rhizoid, petiole, basis, middle, and tip. To reveal the regulatory mechanism of phytohormone synthesis and accumulation in different parts of *S. japonica*, the expression profiles of genes related to phytohormone metabolism were analyzed. As shown in Figure 7B, 14,938 genes were co-expressed in the five parts of *S. japonica*. Based on a functional annotation and a KEGG enrichment analysis, we focused on 127 genes related to phytohormone metabolic pathways and drew an expression clustering heat map based on their FPKM values (Appendix A). We found that *Sjgene 7.85* (encoding an IAA–amido synthetase homolog critical for auxin inactivation) was specifically expressed in the rhizoid and *Sjgene 17.901* (a putative 9-cis-epoxycarotenoid dioxygenase implicated in ABA precursor synthesis) was specifically expressed in the tip. The expression of different genes varied significantly between different parts.

A further analysis demonstrated distinct enrichment patterns of metabolic pathways in intergroup comparisons (Table 4). In the R versus P group, tryptophan metabolism (ko00380) exhibited four DEGs, with one being up-regulated and three down-regulated. In contrast, carotenoid biosynthesis (ko00906) and alpha-linolenic acid metabolism (ko00592) each demonstrated five DEGs, with three being up-regulated and two down-regulated. It is noteworthy that cysteine and methionine metabolism (ko00270) exhibited the most significant differential regulation, with five genes demonstrating an increase and nine genes demonstrating a decrease in expression. In contrast, pyrimidine metabolism (ko00240) exhibited a consistent downregulation in expression for eight genes. When comparing R with B, a divergent profile becomes apparent. Diterpene biosynthesis (ko00904) is characterized by a single up-regulated gene, in stark contrast to the more complex regulation observed in pyrimidine metabolism (ko00240). The latter pathway shows a combination of up-regulated (four genes) and down-regulated (three genes) outcomes among the seven genes analyzed. Cysteine and methionine metabolism (ko00270) exhibited enrichment for a total of 16 DEGs, with 7 up-regulated and 9 down-regulated genes. Similarly, carotenoid biosynthesis (ko00906) and alpha-linolenic acid metabolism (ko00592) each showed a mix of up-regulated and down-regulated genes, with three up-regulated and four down-regulated genes for carotenoid biosynthesis, and three up-regulated and two down-regulated genes for alpha-linolenic acid metabolism. A comparison of groups R and M revealed that pyrimidine metabolism (ko00240) exhibited complete up-regulation (six genes), while DEGs for cysteine and methionine metabolism (ko00270) demonstrated significantly up-regulated expression (nine increased and three decreased). This is in contrast to the balanced regulation of tryptophan metabolism (one increased and one decreased) and carotenoid biosynthesis (three increased and one decreased). In R vs. T comparisons, we found 12 tryptophan metabolism genes (7 increased and 5 decreased), 13 carotenoid biosynthesis genes (8 increased and 5 decreased), and 21 cysteine/methionine metabolism genes (14 increased and 7 decreased). Particularly robust differential expression was observed in pyrimidine metabolism (ko00240), with 16 genes (10 increased and 6 decreased), suggesting systemic nucleotide metabolism alterations.

Intergroup analyses between P-part comparisons showed progressive complexity. The P vs. T group displayed the highest pathway engagement, with cysteine and methionine metabolism reaching 30 differentially expressed genes (20 increased and 10 decreased), while S vs. M comparisons demonstrated complete up-regulation in tryptophan metabolism (7 genes) and diterpenoid biosynthesis (2 genes). Notably, B vs. M comparisons showed minimal pathway involvement, with only three genes enriched in cysteine and methionine metabolism (two increased and one decreased). In contrast, the differential comparisons of the B vs. T and M vs. T groups revealed extensive cross-regulation, particularly in pyrimidine metabolism (nineteen genes each) and carotenoid biosynthesis (seven increased and one decreased, and six increased and two decreased, respectively).

To gain deeper insight into the expression patterns of differential genes within the phytohormone biosynthesis pathway, they were visualized using heatmaps. A comprehensive exploration was undertaken of four phytohormone synthesis pathways (CTK, ethylene, JA, and ABA) that currently possess a higher degree of resolution, with the relevant metabolites partially present in *S. japonica*. These heatmaps were further integrated with the KEGG pathway to construct gene heatmaps for the phytohormone biosynthesis pathway (Figure 9A–D).

Four core biosynthetic genes were identified in the CTK synthesis pathway (Appendix A): 1-deoxy-D-xylulose-5-phosphate synthase (*DXS*); 1-deoxy-D-xylulose-5-phosphate reductoisomerase (*DXR*); isopentenyl-transferases (*IPT*); and cytochrome P450 monooxygenase, family 735, subfamily A (*CYP735A*). *DXS* and *DXR* exhibited peak expression levels in the middle part, whereas *IPT* showed predominant expression in the rhizoid part. *CYP735A*, which encodes the enzyme that catalyzes the hydroxylation of cytokinin precursors to bioactive trans-zeatin, represents a rate-limiting step in cytokinin biosynthesis [34]. Interestingly, the genomic analysis identified ten multicopies of *CYP735A* in *S. japonica* and accumulation in different parts: three multicopies displayed rhizoid-predominant expression, three exhibited tip-specific enrichment, and four showed blade-biased expression profiles.

In the ethylene synthesis pathway, five core biosynthetic genes (Appendix A), golgi transport 1 (*GOT1*), adenylate kinase 1(*AK1*), s-adenosylmethionine synthase (*SAMS*), 1-aminocyclopropane-1-carboxylic acid synthase (*ACS*), and 1-aminocyclopropane-1-carboxylic acid oxidase (*ACO*) were identified, each existing as single-copy or duplicated genes in *S. japonica*. *GOT1* showed maximal expression in the mid-lower blade region, while *AK1* exhibited rhizoid-specific dominance. *SAMS* expression was tip-enriched, and *ACS* demonstrated basal blade preference. Notably, *ACO*, which encodes the terminal enzymatic step in ethylene biosynthesis, displayed dual expression maxima in both the rhizoid and tip parts.

In the JA synthesis pathway, four major synthesis genes were annotated (Appendix A), namely, lipoxygenase (*LOX*), allene oxide synthase (*AOS*), 12-oxo-phytodienoic acid reductase (*OPR*), and acetyl-coA acyltransferase 1 (*ACAA1*), all of which are multicopy genes in *S. japonica*. *LOX* was present in 10 copies in *S. japonica* and was highly expressed mainly in the tips. *AOS* was present in 18 copies in *S. japonica* and showed high expression mainly in the petiole and tips. This enzyme converts LOX-generated 13-hydroperoxyoctadecatrienoic acid (13-HpOTrE) to the unstable propadiene oxide, which is the rate-limiting step in JA synthesis. *OPR* had the highest expression in the basis of *S. japonica*, while ACAA1 was most highly expressed in the rhizoids.

Six major synthetic genes in the two pathways of ABA synthesis in *S. japonica* (Appendix A) were annotated. These include molybdenum cofactor sulfurase (*ABA3*), zeaxanthin epoxidase (*ABA2*/*ZEP*), and xanthoxin dehydrogenase (*ABA2*) in the direct pathway (MVA pathway), as well as the neoxanthin synthase-like enzyme (*ABA4*), 9-cis-epoxycarotenoid dioxygenase (*NCED*), and aldehyde oxidase 3 (*AAO3*) in the indirect pathway (MEP pathway). *ABA1*, *ABA3,* and *ABA4* are single-copy genes, while the remaining three are multi-copy genes. *ABA3* and *ABA1* are involved in the direct pathway of ABA synthesis, with the highest expression found in the basis and tip of the *S. japonica* blade, respectively. *S. japonica* has 15 copies of *ABA2*, which are involved in both direct synthesis and the indirect pathway. Most of the copies are highly expressed in the rhizoids and petioles, while three copies are highly expressed in the tips. *ABA4* is highly expressed in all parts of *S. japonica*, except the tips. The gene encoding *NCED*, the rate-limiting enzyme for ABA synthesis [35] has four copies in *S. japonica*, two of which are highly expressed in the petiole and one in the tip. The gene encoding the final step in ABA synthesis, *AAO3* [36], was also present in *S. japonica* in four copies and was highly expressed mainly in the basis of *S. japonica*, which corresponds to the main site of ABA synthesis.

Finally, using *EF1α* as the internal reference gene, 6 DEGs selected from 127 candidates (Figure 10) were validated. The RT-qPCR analysis demonstrated optimal amplification efficiency, with all reactions exhibiting single-peak melting curves and characteristic sigmoidal amplification profiles, confirming primer specificity and reaction reliability. The expression patterns of these six genes across *S. japonica* parts showed complete concordance with the transcriptomic profiling data, validating the high reproducibility of our sequencing results. This robust technical verification provides a reliable foundation for the subsequent functional characterization of phytohormone metabolic genes in *S. japonica*.

## 4. Discussion

### 4.1. Comparison of Phytohormone Content of S. japonica with Other Algae

In this study, 20 phytohormones (24 in total) were first identified from different parts of *S. japonica* at one time using HPLC-MS/MS. These phytohormones span seven major classes, including auxins, CTK, ABA, GA, SA, ET, and JA. Furthermore, this study identified 12 phytohormones in *S. japonica* for the first time, including GA1 and GA4 from the GA group; MeJA, H2JA, and JA-lie from the JA group; ICA, IPA, IAN, and IAM from the auxin group; trans-zeatin and dihydrozeatin from the CTK group; and the precursor of ethylene synthesis ACC. Among the *S. japonica* phytohormones identified, differences were observed compared to other studies and other algae species.

The auxin (IAA) content in *S. japonica* was detected at 6.802 ng·g^−1^ FW, consistent with the findings of Liu et al. [18]. However, the IAA content of *Laminaria japonica* was 90–95 ng·g^−1^ FW in the study by Li et al. [37] This finding suggests the potential for substantial variations in IAA content among different kelp species. Some differences were found in the IAA content among different algal species, with Phaeophyta having the lowest IAA content, while Rhodophyta had the highest IAA content [38]. Additionally, IAA is not present in some green algae such as *Ulva lactuca* and *Enteromorpha prolifra* [39]. Various types of CTK are found in different algal species. In this study, *S. japonica* showed a total CTK content of 2.868 ng·g^−1^ FW with a 2IP content at 0.172 ng·g-^1^ FW. This was comparable with those of *Enteromor phaprolifra* (0.1 ng·g^−1^ FW) but notably lower than the content of *Pachymenia carnosa* (7.7 ng·g^−1^ FW) and *Chondrus ocellatus* (11.6 ng·g^−1^ FW) [39]. In *S. japonica*, GA analysis identified a GA1 content of 1.204 ng·g^−1^ FW and a GA4 content of 1.042 ng·g^−1^ FW. However, GA3 was not observed, which is typically found in *Enteromor phaprolifra* [39]. The presence of GA in macroalgae is not well reported, but 18–20 GA species were detected in a total of 24 strains of microalgae from the Chlorophyceae, Trebouxiophyceae, Ulvophyceae, and Charophyceae classes, with levels ranging from 342.7 to 4746.1 ng·g^−1^ dry weight (DW) [40].

This study detected 31.786 ng·g^−1^ FW ABA in *S. japonica*, which was significantly higher than those of *Pachymenia carnosa*, *Chondrus ocellatus*, *Ulva Lactuca*, and *Sargassum fusiforme* (22.1, 11.4, 10.4, and 6.1 ng·g^−1^ FW, respectively) [39]. JA has been detected in a variety of lower algal plants, including species from the Chlorophyta (e.g., *Dunaliella* and *Chlorella*), Rhodophyta (e.g., *Gelidium* and *Gracilariopsis*), Euglenophyta (e.g., *Euglena*), and Cyanophyta (e.g., *Spirulina*) [41,42]. The study showed that JA was not detected in seven brown algae, including *Dunaliella dichotoma*, *Ectocarpus fasciculatus*, *Fucus vesiculosus*, and *Saccharina latissimi* [4]. However, in this study, JA was found to be present in *S. japonica* at a level of 405.703 ng·g^−1^ FW. Key enzymes of the JA synthesis pathway, *AOS*, were also found to be present in *S. japonica*, suggesting that *S. japonica* may have the same enzymes of the JA synthesis pathway as are currently present in terrestrial plants but not in *Chondrus ocellatus* [43]. Additionally, the ETH synthesis precursor ACC was detected in macroalgae for the first time, with a content of 13.953 ng·g^−1^ FW in *S. japonica*. Unfortunately, the detection of ETH in *S. japonica* was not achieved under current experimental conditions due to technical limitations associated with its gaseous state instability. It is important to note that no BR was found in *S. japonica* and that the crucial gene for BR synthesis is also missing in *S. japonica*. However, BR is widely present in Chlorophyta [44,45], suggesting that the differences in BR between species are highly significant.

### 4.2. Relationship Between Tissue Specificity and Phytohormone Content in S. japonica

This study revealed significant tissue specificity in the distribution of phytohormones in *S. japonica*. Firstly, the auxins (such as IAA) showed the highest accumulation in the blade (M and T parts) of *S. japonica*, consistent with the findings in a previous study [46]. This finding appears to support the notion of substantial disparities in the structural characteristics of *S. japonica* when compared with terrestrial plants [47,48]. In higher plants, growth hormones are primarily synthesized in meristematic tissues, such as the stem and root tips, and then act through polar transport to other sites to promote cell elongation, root formation, and so on [49,50,51]. In contrast to higher plants, *S. japonica* lacks true root, stem and leaf differentiation, and its growth hormone synthesis does not seem to be confined to a specific region, such as pseudoroots, but rather occurs in the actively growing parts of the foliose thallus. The rapid elongation of *S. japonica* leaves requires growth hormone accumulation at the middle and tips to stimulate cell wall relaxation and longitudinal growth [52].

CTK was relatively less abundant but showed variability among different parts. It was found that CTK expression was highest in the tips of *S. japonica* blades, and the CTK transporter protein ABCG14 was highly expressed in rhizoid [53]. This suggests that the rhizoid may serve as the primary site of CTK synthesis in *S. japonica* (similar to the root tips in higher plants) [54,55]. Highly expressed transporter proteins may be responsible for the active transport of newly synthesized CTK from the rhizoid to other parts. GA exhibited significant positional differences in *S. japonica*, with the highest concentration in the tip and the lowest in the petiole, suggesting a potential role in development and elongation [56,57]. Interestingly, the GA transporter protein NPF3.1 was identified in *S. japonica* [58], which was highly expressed in the rhizoid of *S. japonica* and in the mid-basal part of the blades, and GA transport in *S. japonica* is speculated to be similar to that in higher plants, which is synthesized in the rhizoid and the anterior end of the blades and transported to the tips of the blades to maintain normal growth and division.

ABA content was significantly higher in the tip compared with that in other parts, which is inconsistent with the findings in Wang et al.’s study [46]. This indicates that ABA may play a crucial regulatory role in the tip, potentially related to environmental adaptation and stress responses [59,60]. The elevated ABA levels in the tip could help the cells in this region quickly respond to environmental stresses, such as salinity and temperature, to maintain cellular homeostasis and survival [61,62]. The ABA transporter protein ABCG25 was highly expressed in the rhizoid of *S. japonica* [63], but the ABA content of rhizoid was lower than that of the blades—presumably the rhizoid was responsible for pumping newly synthesized ABA out of the cells and towards the blades via intercellular transport, resulting in a relatively low ABA content in the rhizoid proper. JA demonstrated a highly significant difference with the highest concentration in the tip, indicating an important signaling role in defense responses in this part [64,65]. The JA transporter protein ABCG16 was also highly expressed in *S. japonica* rhizoid [66], and this expression was hypothesized to possibly be the same as that of ABA: rhizoid is responsible for actively pumping newly synthesized JA out of the blade, creating a concentration gradient for transport to the blade, resulting in lower JA content in the rhizoid proper. The ethylene precursor ACC predominantly accumulated in the rhizoid, suggesting its involvement in rhizoid development and environmental stress response [67,68].

Furthermore, the localization patterns exhibited by specific phytohormones were found to be unique. For example, 3-Indoleacetonitrile was exclusively detected in the tips, while SA showed specific accumulation in the rhizoid. These distinct distribution patterns may be linked to the specialized physiological functions of these hormones in these particular regions [69,70], warranting further investigation to elucidate their precise roles.

### 4.3. Tissue Specificity of Phytohormone Biosynthetic Pathway in S. japonica

Phytohormones play a pivotal role in regulating the growth, development, and abiotic stress resistance of *S. japonica* [26]. Understanding how phytohormone synthesis and accumulation are regulated across different parts of *S. japonica* provides valuable insights into its adaptive responses to environmental conditions [71]. This study examined the expression of genes involved in phytohormone metabolism in five different parts of *S. japonica*, revealing tissue-specific patterns consistent with the specialized roles of these regions. The tissue-specific expression of phytohormone biosynthetic genes in *S. japonica* demonstrates the intricate synergistic regulation of phytohormones in different parts [72]. As an organ for fixation and absorption, the rhizoid highly expresses CTK synthesis genes (e.g., *IPT* and some copies of *CYP735A*), ETH synthesis genes *AK1* and *ACO*, and *ABA2* of the direct ABA synthesis pathway, suggesting that it may regulate the growth of other parts through CTK and respond to environmental stresses (e.g., temperature changes) through ETH and ABA [73,74]. The petiole and basis, serving as transitional regions, highly express *AOS* involved in JA synthesis and *NCED* and *AAO3* of the ABA indirect synthesis pathway, which might be involved in JA-mediated defense signaling and ABA-regulated maturation and the abscission of basal cells [35,36,75]. The middle part of the blade, which serves as the primary site of photosynthesis, highly expresses the CTK precursor synthesis genes *DXS/DXR* and the ETH precursor transporter protein *GOT1*, which may promote cell expansion through CTK and coordinate ethylene-mediated leaf senescence [76,77]. On the other hand, the blade tip, known as stress-responsive regions, specifically expresses *CYP735A* (catalytically active tZ-type CTK synthesis), *LOX* (JA synthesis initiation), and *ABA2* and *NCED* for ABA synthesis, suggesting that hormones synergistically regulate tip adaptations to adversity [35,73,78]. These findings indicate the tissue-specific regulation of the phytohormone biosynthesis pathway and provide a molecular-level mechanism for elucidating differences in the growth and environmental adaptation of different parts of *S. japonica*.

### 4.4. Differences in Transcriptional Regulation of DEGs in Different Parts of S. japonica

GO functional annotation and KEGG pathway analysis revealed the tissue-specific metabolic and functional adaptations in different parts of *S. japonica*. By comparing ten pairs of different parts (R vs. P, R vs. B, R vs. M, R vs. T, P vs. B, P vs. M, P vs. T, B vs. M, B vs. T, and M vs. T) and annotating DEGs, this study provides comprehensive insights into the unique roles of each *S. japonica* part. Based on the results of this study and the physiological characteristics of *S. japonica*, the following inferences were made.

**Tissue functional differentiation.** This study reveals a functional differentiation of tissues in different parts of *S. japonica*. The DEGs in the rhizoid and blade are enriched in “carbon fixation” and “carbon metabolism”, indicating that the rhizoid acts as a sequestration organ dependent on the blade for photosynthesis-assimilated products [79,80]. The function of the petiole in *S. japonica* is crucial, and the enrichment of DEGs in the “ribosomal structure” and “nitrogen metabolism” between the petiole and the blade suggests that the petiole acts as a hub connecting the pseudopods to the blade. It is responsible for transporting substances and supporting protein synthesis support functions [81,82,83,84]. Additionally, the presence of “nucleolus” and “preribosome assembly” in P vs. T indicates that petiole cells retain proliferative capacity, while blade tip cells may undergo senescence or differentiation [85,86].

**Photosynthetic gradients.** The photosynthetic organ of *S. japonica* exhibits a gradient regulation, which varies in different regions of the blade. For instance, the DEGs are enriched in “photosystem II” and “chlorophyll binding” at the basis of the blade in comparison with the middle part of the blade. This may be indicative of the higher photosynthetic activity of the basal blade as a nascent region [87], whereas the middle part of the blade is in a mature stage and must maintain homeostasis [88].

**Stress response and defense mechanism.** The tips of the blade were found to be significantly enriched in “proteasome complexes” compared with other parts of *S. japonica* (rhizoid, petiole), probably due to the fact that the tips need to degrade the damaged proteins through the ubiquitin–proteasome system because of the long-term exposure of the tips to bright light, water flushing, or pathogen attacks [89,90,91]. “Peroxisome” pathway enrichment (R vs. B) suggests that rhizoids detoxify ROS generated in hypoxic sediment environments.

**Energy allocation and carbon–nitrogen metabolic coordination.** The widespread enrichment of carbon fixation, the pentose phosphate pathway, and carbon metabolism across comparisons (e.g., R vs. P/B/M, P vs. T) reflects the energy strategies of different parts: the blades are mainly based on prioritizing photosynthetic carbon assimilation, and the rhizoid and petiole may rely on carbohydrate transport and respiration for energy [92,93].

In summary, the integration of GO and KEGG analyses reveals that different parts of *S. japonica* exhibit specialized metabolic and functional roles. The rhizoid is crucial for attachment and nutrient absorption, with significant metabolic activities related to protein phosphorylation and signal transduction. The petioles are involved in structural support. The basis of the blade is specialized in photosynthesis and basic metabolic processes, while the middle part is actively involved in the biological processes of growth, development, and energy storage, and the tips mainly play a role in stress response mechanisms. These findings highlight the complex and coordinated metabolic networks within *S. japonica* tissues, providing a foundation for understanding their roles in growth, development, and environmental adaptation.

## 5. Conclusions

This study systematically analyzed the distribution characteristics of phytohormones in different tissue parts of *S. japonica* and their synthesis and regulation mechanisms by integrating metabolomics and transcriptomics techniques. Firstly, this study revealed significant organizational variability in the distribution of phytohormones. For example, ABA and JA were highly enriched in the blade tips, which may enhance environmental resilience by regulating cellular antioxidant capacity and defense response; SA accumulated only in the rhizoids, presumably related to its fixation and stress tolerance functions. Secondly, the gene expression pattern revealed functional differentiation. The rhizoid, serving as fixation organs, was highly expressed CTK (*IPT*, *CYP735A*) and ETH synthesis genes (*AK1* and *ACO*), suggesting that they regulate overall growth through hormonal signals; blade tips, as stress-responsive regions, specifically expressed JA synthesis genes (*LOX* and *AOS*) and ABA rate-limiting enzyme genes (*NCED*), suggesting that hormones synergistically regulate tip adaptations to adversity. The basal part of the blade displayed enrichment in photosynthesis-related genes, while the middle part was associated with material transport and nitrogen metabolism, reflecting a metabolic division of labor in different regions. Finally, metabolic and transcriptional regulatory networks act in concert. The tissue-specific expression of hormone synthesis genes was highly consistent with their metabolite distribution. For instance, the high expression of *CYP735A* in the rhizoid was associated with CTK accumulation, whereas the expression of *LOX* in the tip of the blades drove the significant enrichment of JA. This regulatory pattern of differential accumulation reflects the evolutionary strategy of *S. japonica* to achieve functional specialization by regulating phytohormone gradients with gene expression networks in the absence of organ differentiation. This study is the first to uncover the distribution characteristics of phytohormones and their synthetic differences in different parts of *S. japonica* and elucidates how *S. japonica* achieves functional specialization through non-specific phytohormone regulation despite lacking organ differentiation, which provides an important theoretical basis for the study of the developmental biology of macroalgae and their mechanism of response to adversity.

## Figures and Tables

**Figure 1 plants-14-01821-f001:**
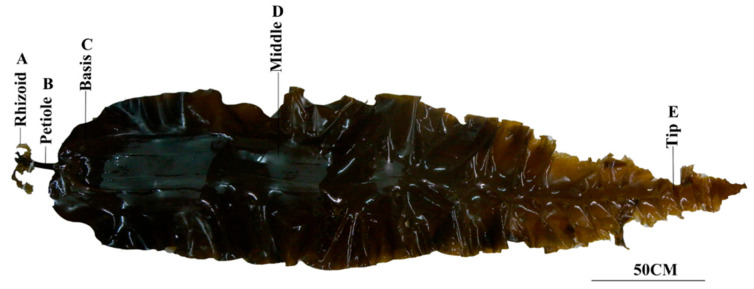
Different parts of *S. japonica*. (A) Rhizoid. (B) Petiole. (C) Basis. (D) Middle. (E) Tip.

**Figure 2 plants-14-01821-f002:**
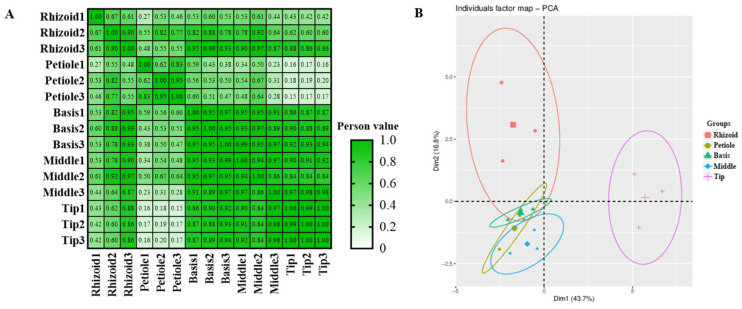
Overview of integrated metabolic data detected in *S. japonica*. (**A**) Pearson correlation of integrated metabolic data. (**B**) Principal component analysis of integrated metabolic data.

**Figure 3 plants-14-01821-f003:**
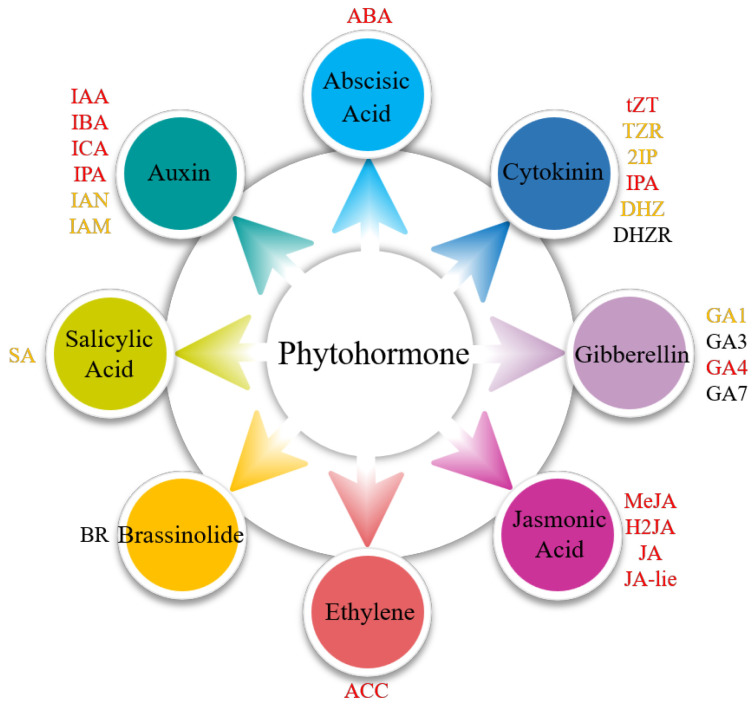
Presence of phytohormones in different parts of *S. japonica*. ABA: abscisic acid. tZT: trans-zeatin. TZR: trans-zeatin-riboside. 2IP: N6-(delta2-isopentenyl) adenine. IPA: N6-(delta2-isopentenyl) adenosine. DHZ: dihydrozeatin. DHZR: dihydrozeatin riboside. GA1: gibberellin A1. GA3: gibberellic A3. GA4: gibberellin A4. GA7: gibberellin A7. MeJA: methyl jasmonate. H2JA: dihydrojasmonic acid. JA: jasmonic acid. JA-lie: jasmonic acid-isoleucine. ACC: aminocyclopropane carboxylic acid. BR: brassinolide. SA: salicylic acid. IAA: indole-3-acetic acid. IBA: 3-indolebutyric acid. ICA: 3-indolecarboxylic acid. IPA: 3-indolepropionic acid. IAN: 3-indoleacetonitrile. IAM: 3-indoleacetamide. The colors indicate the presence of phytohormones as follows: red (detected in the rhizoid, petiole, basis, middle, and tip), yellow (partially detected in five parts), and black (undetectable).

**Figure 4 plants-14-01821-f004:**
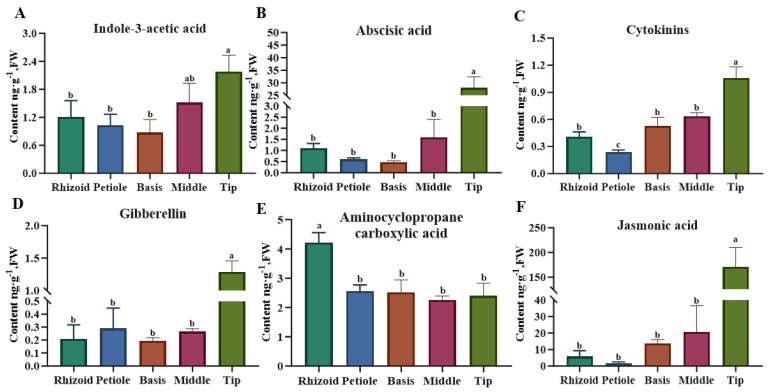
Comparison of the content of major phytohormones in different parts of *S. japonica*. (**A**) Indole-3-acetie acid. (**B**) Abscisic acid. (**C**) Cytokinins. (**D**) Gibberellin. (**E**) Aminocyclopropane carboxylic acid. (**F**) Jasmonic acid. According to Duncan’s multiple range test, different letters indicate significant differences between the means of different parts (*p* < 0.05).

**Figure 5 plants-14-01821-f005:**
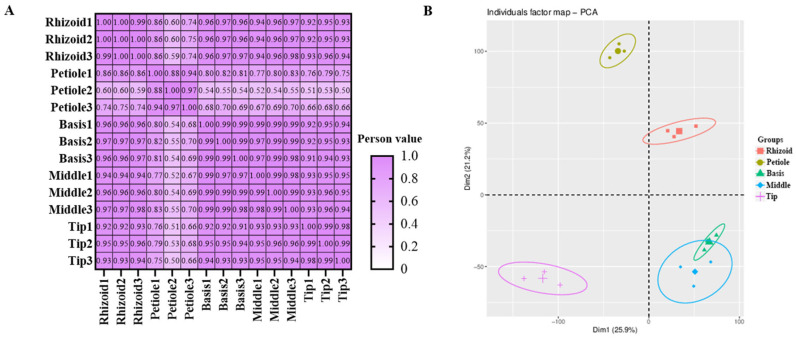
Overview of transcriptomic data detected in *S. japonica*. (**A**) Pearson correlation of transcriptomic data. (**B**) Principal component analysis of transcriptomic data.

**Figure 6 plants-14-01821-f006:**
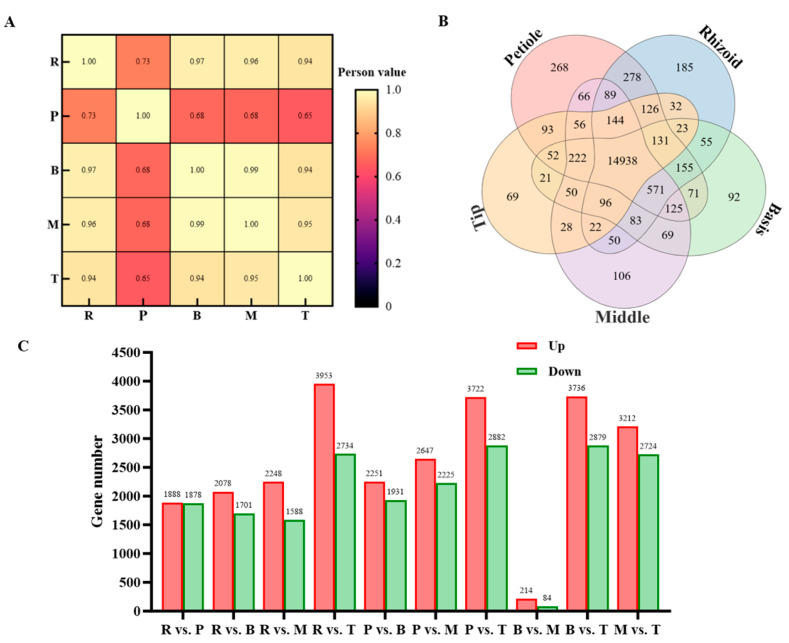
Analysis of DEGs. (**A**) The sample correlation heat map in the R, P, B, M, and T of *S. japonica*. (**B**) Venn diagram of DEGs from different parts of *S. japonica*. The numbers in the figure represent the number of DEGs. (**C**) Number of DEGs from R vs. P, R vs. B, R vs. M, R vs. T, P vs. B, P vs. M, P vs. T, B vs. M, B vs. T, and M vs. T. The numbers in the figure represent the number of DEGs.

**Figure 7 plants-14-01821-f007:**
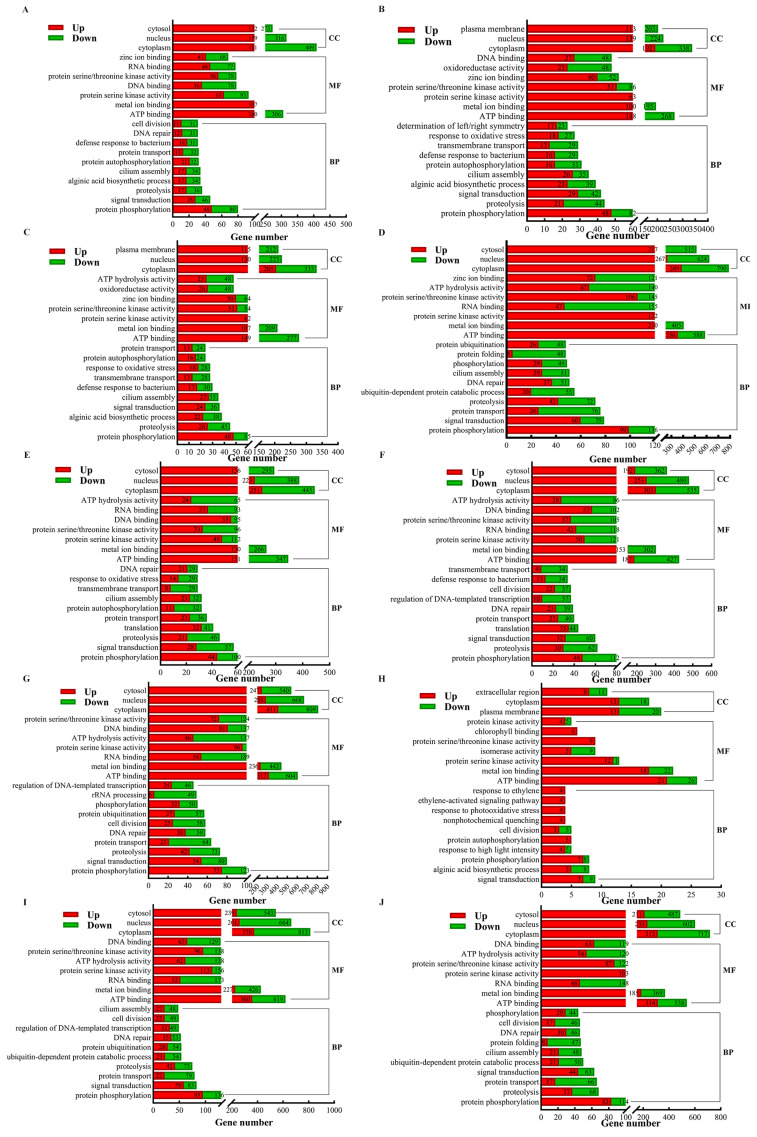
GO annotation classification maps of DEGs from comparison between different parts in *S. japonica*. (**A**) R vs. P. (**B**) R vs. B. (**C**) R vs. M. (**D**) R vs. T. (**E**) P vs. B. (**F**) P vs. M. (**G**) P vs. T. (**H**) B vs. M. (**I**) B vs. T. (**J**) M vs. T. CC: cellular component; MF: molecular function; BP: biological regulation.

**Figure 8 plants-14-01821-f008:**
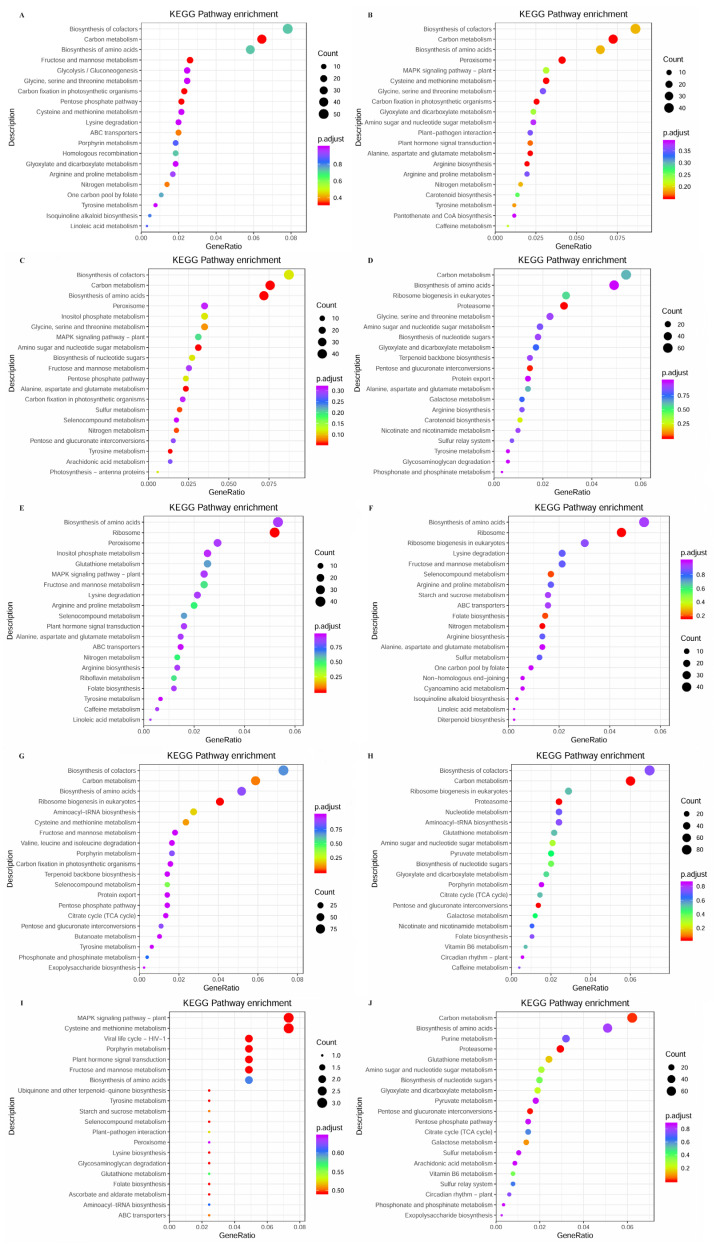
KEGG pathway enrichment analysis of DEGs from comparison between different parts in *S. japonica*. (**A**) R vs. P. (**B**) R vs. B. (**C**) R vs. M. (**D**) R vs. T. (**E**) P vs. B. (**F**) P vs. M. (**G**) P vs. T. (**H**) M vs. T. (**I**) B vs. M. (**J**) B vs. T.

**Figure 9 plants-14-01821-f009:**
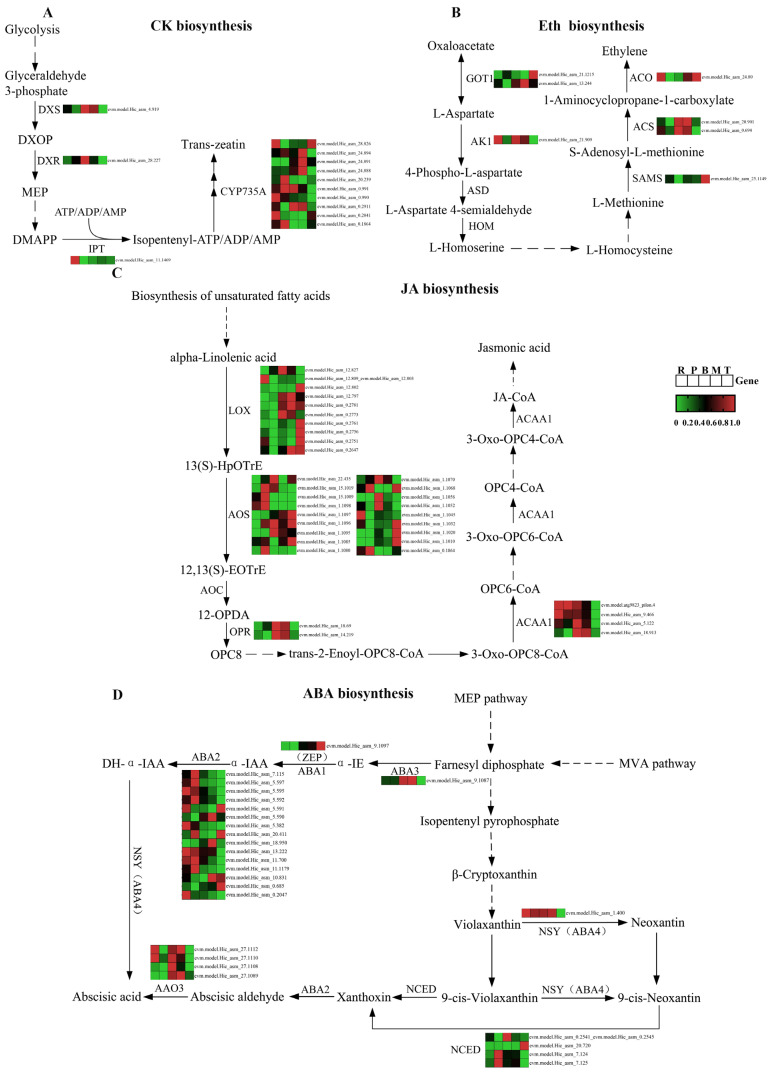
Expression of genes related to mainly phytohormone biosynthesis pathway in *S. japonica*. (**A**) CTK biosynthesis. (**B**) Eth biosynthesis. (**C**) JA biosynthesis. (**D**) ABA biosynthesis.

**Figure 10 plants-14-01821-f010:**
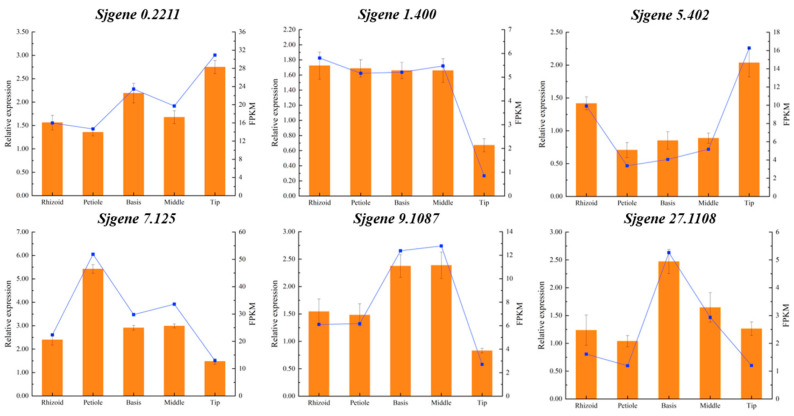
Expression comparison of RT-qPCR and transcriptional analysis of six genes in R, P, B, M, and T from *S. japonica*. The broken line represented gene expression data obtained from a transcriptome sequencing analysis, while the column depicted the gene expression date from RT-qPCR.

**Table 1 plants-14-01821-t001:** Gene primer sequences used for quantitative real-time PCR analysis.

Gene	Forward Primer (5′–3′)	Reverse Primer (5′–3′)
*SjEF1α*	GTGATGGAGGAGAACCC	TTGATGACACCCACAGC
*Sjgene 0.2211*	ACAAGGACGGGAACCACATC	TCACGAACTGATGCGTGTGA
*Sjgene 1.400*	AAGGGAGCTGAAGTGGAACG	GTCCACGTCAAGTGTGTTGC
*Sjgene 5.402*	CCGCGAGTACGACAAGATGA	AGAACGACACCTGCAGACAG
*Sjgene 7.125*	GGCGAATGCGTGTTCATACC	GCTTCGCGTTCATGGTCTTC
*Sjgene 9.1087*	TGTACTTCACGCACTCCACC	AGCCATCTCGTGCTCGTAAG
*Sjgene 27.1108*	CAACATGTACAAGGAGGGCG	CTTGGTCGGAATGACCGAAAG

**Table 2 plants-14-01821-t002:** Detection of targeted metabolome (24 phytohormone content) in different parts of *S. japonica*. According to Duncan’s multiple range test, different letters indicate significant differences between the means of different parts (*p* < 0.05). “NA” represents not detected.

Phytohormone	Test Substances	Parts				
		Rhizoid	Petiole	Basis	Middle	Tip
Auxin	3-Indoleacetamide	0.064 ± 0.009 ^a^	NA	0.040 ± 0.014 ^a^	NA	0.028 ± 0.013 ^a^
	3-Indolecarboxylic Acid	0.456 ± 0.084 ^a^	0.329 ± 0.071 ^ab^	0.324 ± 0.061 ^ab^	0.258 ± 0.019 ^b^	0.336 ± 0.105 ^ab^
	Indole-3-Acetic Acid	1.206 ± 0.288 ^b^	1.028 ± 0.192 ^b^	0.872 ± 0.229 ^b^	1.516 ± 0.337 ^ab^	2.180 ± 0.288 ^a^
	3-Indolepropionic Acid	5.368 ± 0.504 ^a^	5.688 ± 0.419 ^a^	5.686 ± 0.396 ^a^	5.971 ± 0.297 ^a^	5.317 ± 0.836 ^a^
	3-Indolebutyric Acid	1.881 ± 0.157 ^a^	1.978 ± 0.082 ^a^	1.849 ± 0.183 ^a^	1.889 ± 0.197 ^a^	1.518 ± 0.291 ^a^
	3-Indoleacetonitrile	NA	NA	NA	NA	0.185 ± 0.008
Cytokinin	Trans-zeatin	0.116 ± 0.006 ^a^	0.087 ± 0.008 ^b^	0.092 ± 0.018 ^ab^	0.087 ± 0.011 ^b^	0.102 ± 0.031 ^ab^
	Dihydrozeatin	NA	NA	0.052 ± 0.010 ^a^	0.017 ± 0.005 ^b^	0.021 ± 0.012 ^b^
	Trans-zeatin-riboside	NA	NA	NA	NA	NA
	N6-(delta2-Isopentenyl) Adenine	0.045 ± 0.016 ^a^	0.029 ± 0.009 ^a^	0.065 ± 0.025 ^a^	0.033 ± 0.007 ^a^	NA
	N6-(delta2-Isopentenyl) Adenosine	0.246 ± 0.060 ^cd^	0.125 ± 0.018 ^d^	0.280 ± 0.050 ^c^	0.495 ± 0.022 ^b^	0.897 ± 0.053 ^a^
Gibberellins	Gibberellin A1	NA	0.008 ± 0.002 ^c^	0.013 ± 0.007 ^bc^	0.029 ± 0.008 ^b^	1.154 ± 0.112 ^a^
	Gibberellic A3	NA	NA	NA	NA	NA
	Gibberellin A4	0.208 ± 0.089 ^a^	0.282 ± 0.128 ^a^	0.185 ± 0.017 ^a^	0.236 ± 0.025 ^a^	0.131 ± 0.056 ^a^
	Gibberellin A7	NA	NA	NA	NA	NA
Abscisic Acid	Abscisic Acid	1.101 ± 0.174 ^b^	0.602 ± 0.051 ^c^	0.476 ± 0.046 ^d^	1.595 ± 0.662 ^ab^	28.013 ± 3.562 ^a^
Ethylene	Aminocyclopropane Carboxylic Acid	4.215 ± 0.281 ^a^	2.560 ± 0.171 ^b^	2.522 ± 0.343 ^b^	2.255 ± 0.110 ^b^	2.401 ± 0.352 ^b^
Jasmonic Acid	Methyl Jasmonate	3.073 ± 1.188 ^b^	8.224 ± 2.475 ^ab^	5.436 ± 2.338 ^ab^	3.475 ± 0.726 ^b^	9.657 ± 0.580 ^a^
	Jasmonic Acid	5.761 ± 2.891 ^b^	1.865 ± 0.521 ^b^	13.643 ± 1.936 ^b^	20.552 ± 13.119 ^b^	170.665 ± 32.253 ^a^
	Dihydrojasmonic Acid	8.345 ± 2.764 ^b^	2.082 ± 0.973 ^b^	11.206 ± 1.420 ^b^	12.591 ± 2.042 ^b^	128.375 ± 35.707 ^a^
	Jasmonic Acid-Isoleucine	0.129 ± 0.016 ^ab^	0.170 ± 0.026 ^a^	0.159 ± 0.027 ^ab^	0.192 ± 0.028 ^a^	0.104 ± 0.020 ^b^
Brassinosteroids	Brassinolide	NA	NA	NA	NA	NA
Salicylic Acid	Salicylic Acid	9.517 ± 1.486	NA	NA	NA	NA

**Table 3 plants-14-01821-t003:** Quality information on the transcriptome sequencing of *S. japonica*.

Sample	Raw Reads	Clean Reads	Clean Reads Rate (%)	Q30 (%)	GC Content (%)
Rhizoid1	103,336,502	93,647,616	90.62	94.65	54.86
Rhizoid2	99,422,620	90,020,886	90.54	94.72	54.84
Rhizoid3	92,234,828	81,998,948	88.9	94.45	55.03
Petiole1	98,000,366	88,596,096	90.4	94.53	54.97
Petiole2	93,811,592	83,146,322	88.63	94.54	55.29
Petiole3	91,132,630	82,764,530	90.82	94.2	54.83
Basis1	101,323,062	91,874,834	90.68	94.13	55.32
Basis2	90,507,826	81,967,646	90.56	94.56	54.78
Basis3	103,537,970	93,138,382	89.96	94.27	54.87
Middle1	104,956,876	92,055,950	87.71	94.17	55.03
Middle2	103,128,310	93,312,852	90.48	94.25	55.16
Middle3	90,793,274	81,220,748	89.46	93.94	55.06
Tip1	95,509,248	86,173,916	90.23	94.52	55.69
Tip2	96,078,054	87,428,612	91	94.55	56.01
Tip3	91,853,966	82,491,570	89.81	94.5	55.88

**Table 4 plants-14-01821-t004:** Analysis of KEGG pathway related to phytohormone biosynthesis in *S. japonica* transcriptome. The bracketed numbers represent the number of up-regulated genes.

ID	Metabolic Pathway	Gene Number									
		R vs. P	R vs. B	R vs. M	R vs. T	P vs. B	P vs. M	P vs. T	B vs. M	B vs. T	M vs. T
ko00380	Tryptophan metabolism	4 (1)	3 (2)	2 (1)	12 (7)	7 (6)	7 (7)	15 (10)	0	14 (7)	11 (4)
ko00904	Diterpenoid biosynthesis	0	1 (1)	1 (1)	0	1 (1)	2 (2)	0	0	0	0
ko00906	Carotenoid biosynthesis	5 (3)	7 (3)	4 (3)	13 (8)	5 (1)	5 (3)	10 (6)	0	8 (7)	8 (6)
ko00270	Cysteine and methionine metabolism	14 (5)	16 (7)	12 (9)	21 (14)	11 (7)	15 (12)	30 (20)	3 (2)	21 (13)	19 (10)
ko00592	Alpha-Linolenic acid metabolism	5 (3)	5 (3)	6 (4)	6 (3)	3 (2)	5 (3)	7 (4)	0	3 (1)	4 (1)
ko00240	Pyrimidine metabolism	8 (0)	7 (4)	6 (6)	16 (10)	11 (9)	11 (10)	16 (10)	0	19 (8)	19 (8)

## Data Availability

The data presented in this study are openly available from the NCBI. (https://www.ncbi.nlm.nih.gov/bioproject/, reference number PRJNA1244720, accessed on 3 June 2025) and Zenodo (DOI: https://zenodo.org/records/15618312, accessed on 8 June 2025). All original contributions presented in the study are included in the article/Appendix A.

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
