# Peer review of "A Non-Specific Phytohormone Regulatory Network in Saccharina japonica Coordinates Growth and Environmental Adaptation"

_plants, 2025, doi:10.3390/plants14121821_

Round 1
Reviewer 1 Report
Comments and Suggestions for Authors
Line 39-40. You are repeating the information. In the same sentence, you say twice that they are only found in that region.
The information is not organized correctly. For example, line 44 again mentions the production area, when that topic was discussed in the first line. Furthermore, reading the first line does not suggest that its cultivation is that important. Line 48 talks again about its economic importance.
Line 56-64. This information is all about plants. Isn't there any such information on algae? Information from another kingdom is being given. Summarize this information and improve the cohesion of the text.
Line 66. Sargassum sp.?
Line 113. Graphical abstract cannot be in the introduction. Moreover, the figure is too simple.
Line 118. Revise the use of capital letters.
Line 125. “mL” instead of “ml”. Check that this error does not occur again throughout the manuscript.
What is the sense of figure 1?
Section 2.2. Essential information is missing to replicate the study. How was it possible to maintain the sonication temperature so low? What equipment was used?
Line 138-142. Re-write.
English level must be improved thorough all the manuscript.
Line 160. How were they verified?
Line 197. Delete “in this study.”
Line 211. “T)” instead od “T.)”.
Tables/figures should appear the first time they are mentioned in the text, not several pages later. Furthermore, tables/figures mentioned later cannot appear before the table/figure that appears earlier.
Were you able to identify anything in Figure 2a?
Line 222-227. Once again, the information is presented in a disorderly manner.
The figures need to be redone. For starters, most of them are substandard and illegible. Furthermore, they could be reorganized and grouped together to reduce the number of images.
Line 250. “FW”. All abbreviations, no matter how well-known, should be explained the first time they appear in the text. Make sure they are not repeated throughout the manuscript. This is a basic error. The manuscript contains a huge number of basic errors.
Table 2. It would be interesting if this table also included significant differences.
In some parts of the text, first-person verbs appear. A scientific article should never include this type of verb form.
Figure 8. Nothing in the figure is no legible. Furthermore, figures that span multiple pages are not allowed. Consider moving them to supplementary material. Figure 9 is also not legible.
Figure 10. The number of figures in the manuscript is too high. Some need to be moved to supplementary material. Furthermore, the quality of the images needs to be substantially improved.
Comments on the Quality of English LanguageThe English could be improved to more clearly express the research.
Author Response
Thank you for your careful review. Our response is in a word document. Please see the attachment.

Reviewer 2 Report
Comments and Suggestions for Authors
This paper investigates how phytohormones are differentially synthesized in the rhizoid(R),
petiole(P), basis(B), middle(M) and tip(T) to provide deeper insights into the adaptive
mechanisms of S. japonica.
I appreciated the study design but the manuscript is far from being ready to publish. There are some huge issues so I suggest a more careful preparation.
Some issues:
This is wild, please add a REAL Data Availability Statement.
How it currently reads is just MDPI's guidelines and is totally unacceptable. The main value of this work is that these transcriptomics and metabolomics data were obtained. The authors must make the data publically available or the study becomes nearly worthless.
The figure are very blurry. The text should be crystal clear but it is currently difficult to read the fuzzy letters. I can't tell whether the authors or journal portal is at fault - if this is an effect of the manuscript portal than my 'quality of presentation' score should be increased because the figures otherwise look pretty good. Its a shame they were sent to me in such bad resolution.
needs copy editing
Author Response

(The authors gave the same response as above.)

Reviewer 3 Report
Comments and Suggestions for Authors
The manuscript is interesting and have new information, although, there is need to improve the manuscript, mainly, in the material and methods.
Change Ochrophyta to Heterokontophyta
Uniform if is alga (macroalga) or seaweed in the mnauscript
Graphical Abstract need to be added as figure as well and incorporated in the text.
ACN - Please describe the acronym
equpments and model in material and methods
Potential of the sonicator?
C-18 cartridges- model and supplier or producer?
The materials and methods need to be more complete
UPHLC - did the authors run standard samples of phytohormones? or it was based in bibliography? It is a new method ? or it was based in other published article? The method need to be very specific.
Section 2.4, 2.5 and 2.6 needs to renamed, they cannot be the same
Table 2 needs statistical analysis
Author Response

(The authors gave the same response as above.)

Round 2
Reviewer 1 Report
Comments and Suggestions for Authors The authors have made all suggested changesAuthor Response
We are deeply grateful for the considerable time and effort you have devoted to reviewing our paper.
Reviewer 2 Report
Comments and Suggestions for Authors
The authors did not understand nor did they adequately address the feedback from the previous review phase.
Instead of providing a real statement saying where the data was, they simply said that it was in the supplementary Word file or available upon request. This is completely unacceptable for a manuscript that presents transcriptomic profiling and other data-intensive methods. Where is the data? The authors need to host their data in a public repository, it is insufficient to simply accept their word that they did the study without providing any evidence. And by data, here I am referring to the RNAseq reads and expression tables at the very minimum. They should also upload the raw HPLC data as well, but the sequencing reads need to be made available with permanent DOI's or other accessions before the study can be suitable for publication.
Author Response
Please refer to the attached Word document. Thank you.

Reviewer 3 Report
Comments and Suggestions for Authors
The manuscript was well revised, I do not have more revisions questions
Author Response
We are deeply grateful for the considerable time and effort you have devoted to reviewing our paper.
Round 3
Reviewer 2 Report
Comments and Suggestions for Authors
The authors have mostly addressed my comments. Since NCBI can take a lot of time to processes data, I recommend using Zenodo for fast, secure, accessible data hosting.
Author Response
Please see the word, thank you.
